# 17β-Estradiol Can Induce Sex Reversal in Brown Trout

**Jill M. Voorhees** [1,*] , **Elizabeth R. J. M. Mamer** [2], **Daniel J. Schill** [3], **Mitchel Adams** [4], **Carlos Martinez** [4] and **Michael E. Barnes** [1]

1    South Dakota Game, Fish and Parks, McNenny State Fish Hatchery, 19619 Trout Loop, Spearfish, SD 57783, USA
2    ERJMM Research, 15754 Lakeshore Drive, Caldwell, ID 83607, USA
3    Fish Management Solution, 4777 N. Hacienda Ave., Boise, ID 83703, USA
4    United States Fish and Wildlife Service, D.C. Booth Historic National Fish Hatchery, 423 Hatchery Circle, Spearfish, SD 57783, USA
*    Correspondence: jill.voorhees@state.sd.us; Tel.:+1-605-642-6920

**Abstract:** Hormones have been used to change phenotypic sex in many fish species. However, information specific to changing sex in brown trout *Salmo trutta* is lacking. This study compared the effectiveness of two different 17β-estradiol (estradiol) concentrations (20 mg/kg and 30 mg/kg) fed to brown trout for 60 days, beginning at initial feeding. At 456 days post-initial feeding, the 20 mg/kg and 30 mg/kg treatment groups were 84% and 86% female, respectively. These values were significantly higher than the 47% females observed in the control group which did not receive dietary estradiol. At the end of the 60-day estradiol treatment period, weight gain, percent weight gain, and feed conversion ratio were all significantly lower in the tanks of fish receiving estradiol than in the control tanks. Individual fish fed estradiol also weighed significantly less and were significantly shorter than the fish not receiving estradiol-coated feed. Mortality ranged from 1.0 to 2.4% among the treatments and was not significantly different. After 105 days post-initial feeding, weight gain, percent weight gain, and feed conversion ratio were not significantly different among the treatments. At 456 days post-initial feeding, individual fish were significantly longer and heavier in the 20 mg/kg estradiol treatment compared to the fish in the control treatment, and the fish in the 30 mg/kg estradiol treatment were similar to the other two treatments. This study is the first to document the successful sex reversal of brown trout using estradiol. While the estradiol treatments used in this study did not lead to complete feminization, the observed 85-to-15% female-to-male phenotypic ratio indicates the successful feminization of genetic males. The levels of feminization observed in this study are suitable for this initial step to potentially develop a YY male broodstock to control invasive brown trout populations.

**Keywords:** estradiol; feminization; salmonid; brown trout; *Salmo trutta*

## 1. Introduction

Steroids have been used to change the sexual phenotype of over 56 fish species during the past 90 years [1]. Manipulating the sexual phenotype of fish has been used to prevent precocious maturity and reproduction, as well as increase aquacultural production [1,2]. Reversing phenotypic sex using hormones to create single-sex populations and subsequent sterile generations is of interest for fisheries management [1,2]. Recently, using hormones to change genetically-male fish into phenotypical females has become of interest. These sex-reversed fish can be used to produce untreated progeny for release into aquatic ecosystems in an attempt to eradicate undesirable fish populations. This method is known as the Trojan Y Chromosome or YY male approach [3–5]. This approach requires the feminization of genetic XY (or ZZ) and YY males across multiple generations [6–8].

Estradiol is a steroid most commonly used to feminize male fish [1,2]. The inclusion of estradiol in fish diets is the most common delivery method [1,9]. When the hormone is administered through feed, most of it is released into the water in less than 72 h [10],

either via waste or metabolism [2]. Estradiol is typically given to very young fish, with no hormone remaining in the fish several days after cessation of hormone treatments [1,10]. Estradiol treatments are typically given to younger fish because less of the hormone is needed to produce the desired effects. It is important that no hormones remain in the fish that could potentially be consumed.

In addition to dietary inclusion, administered estradiol via immersion and injection have also been used in fish. Immersion uses less hormone compared to dietary inclusion [1,2,9]. However, immersion is much more complex, because a variety of treatment durations and dosages are typically required for fish at different life stages [9]. Immersion also occurs in three forms: continuous [11], discrete [12,13], or as a combination of discrete and dietary treatments [14–18]. Injection is used infrequently because it is arduous and expensive. It also requires more technical skill, carries a risk of secondary infection at the injection site, and is typically only carried out on larger fish [2].

Salmonids are one of the simplest fish families to feminize [1], with 17β-estradiol commonly used [19]. Formulations of different estradiol concentrations and treatment durations have been developed to produce nearly 100 percent phenotypic female populations for several salmonid species, including Atlantic salmon *Salmo salar* [14], brook trout *Salvelinus fontinalis* [7,15], and rainbow trout *Oncorhynchus mykiss* [14]. However, the formulations required to feminize brown trout *Salmo trutta* are unknown. The only published experiment using estradiol to feminize brown trout occurred over 50 years ago [20]. Near-complete mortality was also recorded in this initial study [20]. Such high mortality was likely due to excessive hormone concentrations or treatment duration [1].

Brown trout have been widely introduced beyond their native range, frequently become naturalized, and are considered an invasive species in many locations [21–23]. Eradication using mechanical removal methods, such as electrofishing, has not been successful [24]. Because the YY male approach has shown promise in the control of undesirable brook trout populations [8,25], fisheries managers have been interested in applying these techniques to eradicate invasive brown trout populations. However, to attain an YY male population, effective estradiol concentrations and treatment durations, which do not currently exist, must be developed. Therefore, the objective of this experiment was to evaluate the effects on growth of two potential dietary estradiol formulations on the potential feminization of genetically-male brown trout.

## 2. Methods

On 26 November 2019, McNenny State Fish Hatchery, rural Spearfish, South Dakota, USA received brown trout eggs from Saratoga National Fish Hatchery, Saratoga, Wyoming, USA. The eggs were Plymouth Rock domestic strain brown trout and were at the eyed stage of egg development. The brown trout eggs received were not from a previously sex manipulated population, so the assumption was made that approximately 50% of these fish would be male and approximately 50% would be females. Upon arrival, the eggs were disinfected in 100-ppm active iodine (Ovadine, Syndel Co., Ferndale, WA, USA) for 10 min and then placed into an incubation jar (Eagar Upwelling Incubator Jar, 15.24-cm diameter, Eagar Inc., Randolph, UT, USA). Degassed and aerated well-water was used throughout egg incubation and subsequent experimentation at McNenny State Fish Hatchery (11 °C; total hardness as $CaCO_3$, 360 mg/L; alkalinity as $CaCO_3$, 210 mg/L; pH, 7.6; total dissolved solids, 390 mg/L).

On 5 December 2019, approximately 450 eggs were placed into each of twelve, 2000-L (1.8 m diameter × 0.8 m deep, 0.2 m water depth), circular tanks. Hatching commenced the following day. The experiment began with initial feeding on 30 December 2019 (mean individual fish size ± SE, total length: 23.20 ± 0.19 mm, weight: 0.09 ± 0.00 g, $n = 30$). Three treatments were used, with four tanks per treatment ($n = 4$). The 60-day treatments varied by dietary 17β-estradiol (hereafter referred to simply as estradiol) concentration: (1) 0 mg estradiol/kg of feed, (2) 20 mg estradiol/kg of feed, and (3) 30 mg estradiol/kg of feed (hereafter referred to simply as mg/kg). The dosages were based on the concentrations

that proved to be successful in changing the phenotypic sex in other studies with salmonids, in addition to trying to minimize the amount of hormone used.

Estradiol was diluted with ethanol at a 4:1 ratio and applied to the feed using a continuous spray pump food mister to ensure uniform coverage. Ethanol (without estradiol) was also applied to the feed for the control group. One mister was used for each treatment to avoid possible cross-contamination. Administration of estradiol-treated feed began at initial feeding and continued for the next 60 days.

Each tank of fish received 50 g/day of feed beginning at initial feeding, and continuing for four days. Beginning on the fifth day of feeding, feeding rates were determined by the hatchery constant method [26]. Based on prior experience with brown trout rearing at McNenny State Fish Hatchery, a projected growth rate of 0.055 cm/day and 1.1 feed conversion ratio was used for the next 56 days. Automatic feeders dispensed feed every 15 min during daylight hours. After the 60-day estradiol treatment period, projected growth rates, used to calculate feed amounts, were increased to 0.057 cm/day. BioVita Starter (Bio-Oregon, Longview, Washington, DC, USA) size #0 was used for the first 30 days of feeding. BioVita Starter size #1 was used for the next 30 days, and BioVita Starter #2 was used for the next 40 days. Protec 1.5 mm (Skretting USA, Tooele, UT, USA) feed was fed for the remaining 100 days of rearing at McNenny State Fish Hatchery.

All the fish in the tanks were weighed, to the nearest 0.01 kg, at the end of the 60-day estradiol treatment period. In addition, 10 fish per tank (40 fish per treatment) were anesthetized using 60 mg/L tricaine methanesulfonate (MS-222; Tricaine-S, Syndel, Ferndale, WA, USA), weighed to the nearest 0.01 g and measured (total length) to the nearest 0.01 mm. At the end of the 60-day period, the fish were moved because of changes in tank availability at McNenny State Fish Hatchery. All of the fish in each respective tank were moved to a 180-L circular tank for the remainder of rearing at McNenny State Fish Hatchery. No commingling of trout occurred during this process; all of the fish in a larger tank were just moved into a smaller tank. At the end of 105 days of rearing, all fish in each tank were weighed to the nearest 0.01 kg to obtain rearing data for the final time. Five fish from each tank were anesthetized using 60 mg/L MS-222 (20 fish per treatment), weighed to the nearest 0.01 g, and measured (total length) to the nearest 0.01 mm. To maintain acceptable rearing densities, half of the fish in each tank, at random, were selected to be culled. On rearing day 170, 75 fish from each tank (300 from each treatment) were anesthetized using 60 mg/L MS-222, weighed to the nearest 0.1 g, measured (total length) to the nearest 1.0 mm, and tagged with a passive integrated transponder (PIT) tag (Biomark MiniHPT10 PIT Tag, Biomark, Inc., Boise, ID, USA). After tagging, all fish were placed into one communal 2000-L tank for 27 days until they were transported to the D.C. Booth National Historic Hatchery (Spearfish, SD, USA) for an additional 286 days of rearing until the fish were at a size large enough to accurately determine phenotypic sex.

After 456 days of rearing, 240 fish from each treatment (60 fish from each initial tank) were euthanized using a lethal MS-222 dose of 200 mg/L. These fish were weighed to the nearest 0.1 g and measured (total length) to the nearest 1.0 mm. Phenotypic sex was then determined by gonadal examination during necropsy. At this size, it was possible to determine if a fish had testes or developing egg sacs. There were no observed intersex fish in any of the fish that were examined.

This experiment was performed within the guidelines set out by the Aquatics Section Research Ethics Committee of the South Dakota Game, Fish and Parks (approval code, SDGFPARC20201) and within the guidelines for the Use of Fishes in Research set by the American Fisheries Society.

Data were analyzed using the SPSS statistical program (24.0, IBM, Armonk, NY, USA). Phenotypic sex frequency data (%) and percent mortality were log transformed prior to data analysis [27]. One-way Analysis of Variance (ANOVA) was performed on all variables and if significant differences were observed, Tukey's post-hoc means testing procedure was conducted. Because the tanks were the experimental unit (and not the fish), the length and

weight data from individual fish was first averaged by tank, and then this average was used in the one-way ANOVA. Significance was pre-determined at $p < 0.05$.

## 3. Results

A significantly greater percentage of phenotypic females was observed in the 20 mg/kg and 30 mg/kg treatments compared to the control, 0 mg/kg, treatment (Figure 1). However, there was no significant difference in the percentage of phenotypic females between the two estradiol treatments, with 84% and 86% females in the 20 mg/kg and 30 mg/kg treatments, respectively. Both of these values were significantly higher than the 47% females observed in the control treatment.

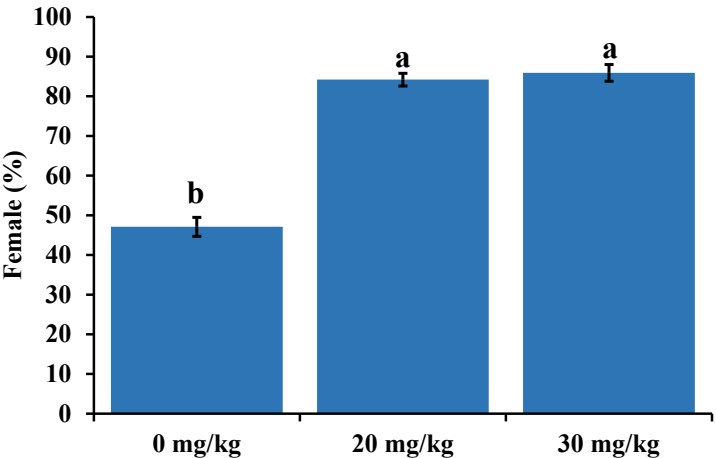

**Figure 1.** Mean (±SE) percentage of females in groups of brown trout *Salmo trutta* fed three different concentrations of estradiol/kg of feed for 60 days, starting at initial feeding. Bars with different letters above differ significantly ($p < 0.001$).

At the end of the 60-day estradiol treatment period, total weight gain, percent weight gain, and feed conversion ratio were all significantly greater in the control tanks than either the 20 mg/kg or 30 mg/kg estradiol treatments (Table 1). Mortality ranged from 1.0 to 2.4% among the treatments and was not significantly different. After 105 days, weight gain, percent weight gain, and feed conversion ratio were not significantly different among the treatments.

**Table 1.** Mean (±SE) total weight gain, percent weight gain, feed conversion ratio (FCR [a]), and mortality for brown trout *Salmo trutta* fed three different concentrations of estradiol/kg of feed for 60 days starting at initial feeding. Means with different letters in the same row differ significantly ($p < 0.05$, $n = 4$).

| Rearing Day | | Estradiol (mg/kg) | | | |
| --- | --- | --- | --- | --- | --- |
| | | **0** | **20** | **30** | *p*-**Value** |
| 60 | Gain (kg) | 0.59 ± 0.02 z | 0.44 ± 0.02 y | 0.42 ± 0.02 y | <0.001 |
| | Gain (%) | 1465 ± 42 z | 1103 ± 54 y | 1053 ± 55 y | <0.001 |
| | FCR | 1.43 ± 0.04 z | 1.91 ± 0.09 y | 2.01 ± 0.10 y | <0.001 |
| | Mortality (%) | 1.0 ± 0.6 | 1.4 ± 0.2 | 2.4 ± 0.7 | 0.08 |
| 105 | Gain (kg) | 0.53 ± 0.02 | 0.49 ± 0.04 | 0.44 ± 0.02 | 0.25 |
| | Gain (%) | 84 ± 5 | 101 ± 9 | 97 ± 7 | 0.27 |
| | FCR | 2.65 ± 0.11 | 2.94 ± 0.31 | 3.15 ± 0.17 | 0.31 |

[a] FCR = food fed/weight gained.

After the first 60 days of the study, individual fish were significantly heavier and longer in the control tanks (Table 2). In addition, the fish receiving 20 mg/kg estradiol were heavier and longer than those receiving 30 mg/kg.

**Table 2.** Mean (±SE) fish length and weight for brown trout *Salmo trutta* fed three different concentrations of estradiol/kg of feed for 60 days starting at initial feeding. Means with different letters in the same row differ significantly ($p < 0.05$, $n = 4$).

| Rearing Day | | Estradiol (mg/kg) | | | *p*-Value |
|---|---|---|---|---|---|
| | | 0 | 20 | 30 | |
| 60 | Length (mm) | 52.28 ± 0.20 z | 46.12 ± 0.31 y | 43.53 ± 0.62 x | <0.001 |
| | Weight (g) | 1.46 ± 0.01 z | 1.07 ± 0.02 y | 0.96 ± 0.05 y | <0.001 |
| 105 | Length (mm) | 65.86 ± 1.22 | 61.99 ± 2.36 | 60.98 ± 0.79 | 0.13 |
| | Weight (g) | 2.86 ± 0.12 | 2.29 ± 0.25 | 2.42 ± 0.12 | 0.10 |
| 170 | Length (mm) | 95 ± 1 z | 92 ± 0 y | 89 ± 1 x | <0.001 |
| | Weight (g) | 9.7 ± 0.2 z | 9.1 ± 0.1 y | 8.2 ± 0.2 x | <0.001 |
| 456 | Length (mm) | 188 ± 1 y | 201 ± 8 z | 191 ± 1 zy | 0.03 |
| | Weight (g) | 69.8 ± 1.3 y | 79.7 ± 3.5 z | 72.2 ± 1.3 zy | 0.03 |

There were no significant differences in weight or total length among the treatments at 105 days. At day 170, the control fish were once again heavier and longer than the fish which had received estradiol, with the 20 mg/kg treatment fish were also heavier and longer than the 30 mg/kg fish. At 456 days, fish in the 20 mg/kg estradiol treatment were significantly longer and heavier than the fish in the control treatment.

## 4. Discussion

This study is the first to document the successful phenotypic sex reversal of brown trout using estradiol. It should be mentioned that the genotypic sex of any of these fish were never actually determined. However, with the assumption that the sex ratio would be close to 50/50 male-to-female ratio, and with the fish in the control treatment being approximately 50%, our speculation was validated. These results indicate that 20 mg/kg for the first 60 days of feeding is just as effective as 30 mg/kg in feminizing male brown trout. However, while both formulations had a similar proportion of female brown trout, neither concentration produced the total, or near-total, sex reversal that has been observed in other salmonids. Feeding brook trout 20 mg/kg of feed for 60 days led to 99% of the study population being phenotypic females [7,15]. Johnstone et al. [14] was able to produce 100% females in Atlantic salmon and rainbow trout by feeding 20 mg/kg of estradiol for only 30 days. In addition, dietary estradiol has been used with several other non-salmonids, with nearly 100% phenotypic female sex ratios observed in channel catfish *Ictalurus punctatus* [28], convict cichlid *Amatitlania nigrofasciata* [29], and pejerrey *Odonthesthes bondariensis* [30].

Although salmonids can be relatively easy to feminize, there can be notable differences for even closely related species [1]. Thus, the results of the present study may be unique to brown trout and may also be unique to the strain of brown trout used. The estradiol concentrations used in this study likely also influenced the results. Formulations using either higher estradiol amounts, or longer durations, may be needed to acquire greater phenotypic sex reversal. In other fish species, relatively small changes in estradiol concentrations have led to changes in the efficacy of sex reversal. For example, 150 mg/kg of estradiol for 60 days was required to produce 100% female populations of bluegill *Lepomis macrochirus*, with concentrations below 150 mg/kg estradiol producing intersex fish [31].

The decreased growth in brown trout size after receiving estradiol-containing feed has been observed in other species. Bluegill, rainbow trout, Atlantic salmon, and many other fish species have also been observed to experience decreased initial growth when consuming feed containing estradiol [14,19,31]. Similar to the present study, an initial period of slower growth, followed by relatively rapid compensatory growth after the cessation of estradiol treatment was also reported by Johnstone et al. [14], Feist et al. [19], Wang et al. [31], and Schill et al. [7]. The reason for the decrease in initial growth associated with estradiol feeding is unknown. The decreased growth during the first 60-days could be due to palatability issues associated with estradiol. However, numerous studies report

similar initial growth issues, while compensatory growth observed in this study is less frequently observed [31]. In contrast, some studies have even documented increased growth associated with estradiol-treated feed in yellow perch *Perca flavescens* and also Japanese eel *Anguilla japonica* [32,33].

The feed conversion ratio in this study after 60 days is typical for brown trout. The ratio of 1.43 from the control tanks is similar to the ratio of 1.58 observed in brown trout observed at McNenny State Fish Hatchery in 2010, the most recent year that brown trout were grown at a production scale at the hatchery [34]. The higher feed conversions in the tanks receiving estradiol likely indicates poor palatability [35,36], although palatability was not directly measured. The feed conversion ratios of over 2.5 in all treatments at 105 days was likely a result of overfeeding. These ratios are much higher than that reported previously for brown trout after initial feeding [37,38], except for those studies where the fish were fed to slightly above satiation on experimental diets [39,40].

Because brown trout survival in this study during and after estradiol treatment was high and similar among the treatments, the formulations used did not appear to cause any detectable health issues, at least at the levels sufficient to induce mortality. These results are similar to Wang et al. [31], who also observed no estradiol-induced mortality. However, other studies have reported increased mortality for fish fed hormone-treated feed [1,41,42]. The difference among the studies is likely due to differences in estradiol concentrations, with higher levels causing mortality in fish fed treated feed [1].

While the estradiol treatments used in this study did not lead to complete feminization, the observed 85-to-15% female-to-male phenotypic ratios for both treatment regimens indicate the successful feminization of genetic males. The observed levels of feminization resulting from either of the estradiol treatments is sufficient for the future development of a YY male broodstock [7]. The information in this study describing the sex-reversal of male brown trout, coupled with the recently developed genetic sex marker for brown trout [43], makes the next phase of YY male broodstock development possible. The genetic sex marker has an average concordance rate of 96% ($n = 688$) between observed phenotype and marker-predicted genetic sex for individual waters in the five western states in the United States where it has been tested [43]. The genetic sex marker would have made examining the genetic sex of this fish possible; however, the sex marker was not yet finalized when this study was performed. This next step in applying this research to the extirpation of invasive brown trout populations involves spawning feminized XY or FXY individuals with fish in an unrelated group of regular XY males to produce YY males [7]. However, it is possible that this cross between FXY females and XY males may not produce viable progeny, as has been observed in other fish species [7]. If these fish can produce viable embryos, then the progeny from these families would be 25% XX, 50% XY, and 25% YY males. Further genetic data analysis would enable differentiation between XY and YY males in this cohort, with subsequent feminization of half of the YY fish during the final phase of YY brown trout broodstock development [7,16]. After this final development occurs, progeny from these broodstock fish could then be stocked into wild populations. Subsequent generations of YY males stocking would cause an ever greater number of XY males, which would eventually skew the wild population to an all-male population [7]. Pairing this skewing toward males with the physical removal of fish from the wild population could lead to an even quicker achievement of a complete male population [7]. However, complete eradication could only be achieved in a closed system, where no new wild XY males could enter the system.

## 5. Conclusions

In conclusion, although male brown trout appear to be less susceptible to feminization than other salmonid species, the results of this study indicate that they can still be feminized at relatively high levels. This is the first study to document the successful sex reversal of brown trout using estradiol. This experiment lays the foundation for future research to determine the formulation, estradiol concentration, and treatment time needed to reach complete sex reversal in brown trout. However, even at the less-than-complete

feminization rates observed in this study, estradiol dosages and the treatment durations used make it possible to further evaluate the use of YY male to eradicate invasive brown trout populations.

**Author Contributions:** Conceptualization: J.M.V., M.E.B., E.R.J.M.M. and D.J.S.; Methodology: J.M.V. and M.E.B.; Data Curation: J.M.V., E.R.J.M.M. and D.J.S.; Formal Analysis: J.M.V. and M.E.B.; Investigation: J.M.V., M.E.B., E.R.J.M.M., D.J.S., M.A. and C.M.; Project Administration: J.M.V. and M.E.B.; Resources: J.M.V., M.E.B., E.R.J.M.M. and D.J.S.; Supervision: MEB; Roles/Writing—original draft: J.M.V.; Writing—review & editing: J.M.V., M.E.B., D.J.S. and C.M. All authors have read and agreed to the published version of the manuscript.

**Funding:** This research received no external funding.

**Institutional Review Board Statement:** This experiment was performed within the guidelines set out by the Aquatics Section Research Ethics Committee of the South Dakota Game, Fish and Parks (approval code, SDGFPARC20201) and within the guidelines for the Use of Fishes in Research set by the American Fisheries Society.

**Data Availability Statement:** Data available upon request.

**Acknowledgments:** We thank Carson Voorhees, Kenton Voorhees, Jeremy Kientz, Edgar Meza, Nathan Huysman, Eric Krebs, Benj Morris, Lauren Van Rysselberge, Lynn Slama, and Michael Roubidoux for their assistance with this study.

**Conflicts of Interest:** The authors declare no conflict of interest.

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
