# Peer review of "17β-Estradiol Can Induce Sex Reversal in Brown Trout"

_fishes, doi:10.3390/fishes8020103_

Round 1
Reviewer 1 Report
In this paper, the authors report on the efficacy of estradiol to induce sex reversal in brown trout. The study offers but few new insights into effects of exogenous steroids on sex determination / sex ratios, and hence, this article can be shortened. The experimental design is technically sound, but the choice for 20 and 30 mg estradiol / kg feed is so close that the failure to detect a difference was to be expected, and it is not clear why the concentration of the second experimental group was not changed in a 3-5 fold step. The biggest issue, in reviewer’s opinion, is that no confirmation is provided anywhere that the experimental fish “were genetically-male brown trout”. Authors are commended for the use of replicate tanks, data from which need to be analysed differently (nested design). Please note the not insignificant number of grammatical errors in the text.
Title: runs better if a verb and the species name are included in the title, e.g. “17beta-estradiol can induce sex reversal in brown trout, Salmo trutta”
L18: reviewer argues that “…weight gain, percent weight gain, and feed conversion ratio…” “…were lower in fish from the tanks receiving…” (i.e, add “in fish”).
L19: ‘…weight and length… were …smaller…” (not “less”)
L42” ‘dietarily’ need re-wording to a more accepted term
L49: ‘…as a combination of…’
L53: add ‘beta’ symbol to estradiol
L70: how do the authors know that the trout they used were genetically male? This is a key statement, that defines the results, but no evidence for the correctness of this statement is provided; to confirm they are phenotypically male is easy, but genetically…? This needs addressing in the Discussion.
L82: sentence reads easier if stating ‘twelve’
L83: ‘commenced’
L86: authors are commended for the use of replicate tanks, something that is often not done for lack of resources or that is overlooked
L88: the experimental design is not great; 20 or 30 mg/kg feed are sufficiently close together, that a priori, a difference in effect between these would be unlikely. At least a tripling (c.f. 1/3) or quadrupling (1/4) in concentrations would have been much more insightful.
L105, L111: please explain how the fish were weighed and measured – was anaesthetic used? If not, how would the fish keep still to allow length estimates to be made? How can measurements be made to the nearest 0.01 mm, and is reporting to that level appropriate? That seems like an impossible task, unless this is done by micrometer eyepiece through a microscope….
L109: ‘…a…tanks…’?
L110: ‘commingling’?
L112: ‘weighed’
L115:were the fish ‘randomly culled’ or were ‘random fish culled’?
L132; please explain which response variables were analysed using ANOVA. Please note that this is nested design and that averaging the data from each tank for a single value per tank ends up resulting in loss of depth of data; please consider running the analysis as a nested ANOVA. Further: as % data are not normally distributed, they must be analysed with non-parametric chi-square or Fisher’s exact test.
L140: ‘was’
Figure 1: it is not clear why authors use ‘y’ and ‘z’ above bars. Lettering is arbitrary, but it makes sense to run with convention, and use ‘a’ and ‘b’ instead. Please note that the ‘z’ above 30 mg/kg is too close to the bar. Please rephrase the figure legend: “Percentage of female trout in response to feeding ….”
Table 1: why are Gain (kg) and next part of the remainder of that line in bold font?
L162: please delete ‘also’; this is already covered by ‘in addition’, earlier in the sentence.
Table 2: why are length (mm) and next part of the remainder of that line in bold font? Should Rearing Day 90 be reading Day 170? It is not in chronological order…
L178: ‘salmon’
Discussion: authors comment (L183) on variability between closely related species; they then suggest that estradiol concentrations may come into play – after previously making direct comparisons with related species exposed to the SAME estradiol concentrations. This is a contradiction in terms. Rather, the authors might want to consider possible effects of temperature, which is a known factor to affect sex ratios, also in salmonids (there are numerous papers reporting on this).A nother explanation is that the stock were not all genetic male – this should have been tested, if not for all, then at least for a proportion (a sample) of the experimental fish.
L202: I struggle with estradiol affecting palatability, and the listed citations do not seem to relate to estradiol per se. A stronger case would be needed to associate the effect with palatability; changes in physiology seem a more likely explanation in reviewer’s opinion.
Author Response
Comments are in attached word document.

Reviewer 2 Report
The manuscript entitled “17β-estradiol Sex Reversal of Brown Trout ” is the first study to report that, the sex of brown trout could be reversed by dietary estradiol. In this manuscript they also analyzed the effects of dietary estradiol on the growth performance of brown trout. Although the manuscript design is simple and its content is relatively few, it gives some implications for the establishment of mono-sex production technology in this species to prevent its invasion. My comments are listed below.
1. The title should be modified. It should better provide detailed information about your own experiments or findings.
2. How do you know the sex determination system of your brown trout is XX-XY? The fish sex determination system is very plastic. Can you find the genetic sex marker mentioned in reference 41? Can you use the sex-linked marker of brown trout to confirm the genotype of the possible sex-reversed fish?
3. The description of fish culture is a little cumbersome for you have transferred the fish to different places. You may use a table to summarize the different culturing stages.
4. How the phenotypic sex was determined by gonadal examination during necropsy? The method for judging sex should be described. Some gonadal photos should be added. It is better to determine the phenotypic sex by HE staining. Intersex is very common in hormone-treated fish. Did you find intersex fish?
5. In Table 2, are the data of 90 days and 105 days reversed?
6. If you want to calculate the feed conversion ratio accurately, you should make sure that the uneaten feed were weighted. I did not find such descriptions in the M&M. And it is almost impossible to measure the weight of the uneaten tiny feed.
Author Response
Comments are in attached document.

Round 2
Reviewer 2 Report
The authors have addressed well all my questions and this manuscript is now acceptable for publication.
Author Response
Thank you.